# The Relation between Hair-Cortisol Concentration and Various Welfare Assessments of Dutch Dairy Farms

**DOI:** 10.3390/ani11030821

**Published:** 2021-03-15

**Authors:** Frank J. C. M. van Eerdenburg, Tessa Hof, Benthe Doeve, Lars Ravesloot, Elly C. Zeinstra, Rebecca E. Nordquist, Franz Josef van der Staay

**Affiliations:** 1Department of Population Health Sciences, Division of Farm animal Health, Faculty of Veterinary Medicine, Utrecht University, Yalelaan 7, 3584 CL Utrecht, The Netherlands; 2Behaviour and Welfare Group, Faculty of Veterinary Medicine, Utrecht University, Yalelaan 7, 3584 CL Utrecht, The Netherlands; t.hof518@gmail.com (T.H.); benthedoeve@gmail.com (B.D.); lars.ravesloot@wur.nl (L.R.); e.c.zeinstra@uu.nl (E.C.Z.); r.e.nordquist1@uu.nl (R.E.N.); f.j.vanderstaay@uu.nl (F.J.v.d.S.)

**Keywords:** hair cortisol, dairy cattle, welfare assessment, stress, health

## Abstract

**Simple Summary:**

Many protocols have been developed to assess farm animal welfare. However, the validity of these protocols is still subject to debate. The present study aimed to compare eight welfare assessment protocols. Chronic stress has a negative impact on animal welfare and causes an increase in cortisol, which can be objectively measured in hair. Hair cortisol concentration has been suggested as reflecting the stress level over a long period of time. Correlation coefficients were calculated between each of the welfare assessment protocol scores and mean hair cortisol concentrations from 10 cows from 58 dairy farms spread over the Netherlands. We expected a negative correlation between cortisol and the result of the welfare protocol scores. However, most protocols or components were uncorrelated with hair cortisol and we did not find evidence supporting our hypothesis. This suggests that the protocols might not yield valid indices for cow welfare, or alternatively, that hair cortisol levels may not provide a long term indicator for stress in dairy cattle.

**Abstract:**

Many protocols have been developed to assess farm animal welfare. However, the validity of these protocols is still subject to debate. The present study aimed to compare nine welfare assessment protocols, namely: (1) Welfare Quality^©^ (WQ), (2) a modified version of Welfare Quality (WQ Mod), which has a better discriminative power, (3) WelzijnsWijzer (Welfare Indicator; WW), (4) a new Welfare Monitor (WM), (5) Continue Welzijns Monitor (Continuous Welfare Monitor; CWM), (6) KoeKompas (Cow Compass; KK), (7) Cow Comfort Scoring System (CCSS), (8) Stall Standing Index (SSI) and (9) a Welfare Index (WI Tuyttens). In addition, a simple welfare estimation by veterinarians (Estimate vets, EV) was added. Rank correlation coefficients were calculated between each of the welfare assessment protocol scores and mean hair cortisol concentrations from 10 cows at 58 dairy farms spread over the Netherlands. Because it has been suggested that the hair cortisol level is related to stress, experienced over a long period of time, we expected a negative correlation between cortisol and the result of the welfare protocol scores. Only the simple welfare estimation by veterinarians (EV) (ρ = −0.28) had a poor, but significant, negative correlation with hair cortisol. This correlations, however, failed to reach significance after correction of *p*-values for multiple correlations. Most of the results of the different welfare assessment protocols had a poor, fair or strong positive correlation with each other, supporting the notion that they measure something similar. Additional analyses revealed that the modified Welfare Quality protocol parameters housing (ρ = −0.30), the new Welfare Monitor (WM) parameter health (ρ = −0.33), and milk yield (ρ = −0.33) showed negative correlations with cortisol. We conclude that because only five out of all the parameter scores from the welfare assessment protocols showed a negative, albeit weak, correlation with cortisol, hair cortisol levels may not provide a long term indicator for stress in dairy cattle, or alternatively, that the protocols might not yield valid indices for cow welfare.

## 1. Introduction

Although there is no specific EU directive for dairy cows, a recent report by the EU Parliament’s Directorate-General for Internal Policies stated that “dairy cow welfare may be considered to be the second greatest animal welfare problem in the EU” [1,2]. Accurate and frequent welfare assessment is the first step towards improvement of cow welfare on the farm. However, a ‘gold standard’ for welfare assessment is still lacking. Defining animal welfare for dairy cows is heavily influenced by moral and ethical standards of the society. One may argue that objectivity is impossible regarding measuring animal welfare because it is based on what is socially regarded as ‘acceptable’ [3]. However, one may still strive for objectivity within an ethical consensus. An animal welfare assessment protocol, Welfare Quality^®^ (WQ) [4], was developed based on the concept of the Five Freedoms [5,6]. However, the validity of the aggregation of all measured parameters in the WQ protocol as index of welfare has been disputed [7]. Furthermore, WQ was found not to be very discriminative as nearly all farms were classified as ‘acceptable’ in several studies [8,9,10]. Van Eerdenburg et al. modified WQ to increase the discriminative capacity (WQ-Mod) [11]. In this same study, WQ and WQ-Mod were correlated with other Dutch welfare assessment protocols for dairy cattle; KoeKompas (=Cow Compass) (KK), WelzijnsWijzer (= Welfare Indicator) (WW) and Continue Welzijns Monitor (=Continuous Welfare Monitor) (CWM) (See Table 1 and Table 2). Unfortunately, these protocols are only available in the Dutch language. KK is based on the ‘common sense’ of the makers, but largely includes the Five Freedoms in its criteria [12]. WW is also based on the Five Freedoms [13]. CWM is a monitor based solely on readily available recorded data about a farm, such as the number of deaths, but also includes economic features like milk yield [14]. In addition to these protocols, the present study includes the Cow Comfort Scoring System (CCSS), that is mainly based on environmental parameters [15]. Furthermore, Tuyttens et al. developed a Welfare Index (WI Tuyttens), based on expert opinions and data from the WQ protocol [16]. Finally, a new welfare monitor (WM) was designed, based on the modified WQ protocol, but executable in around 1 h [17]. All aforementioned assessment protocols are large and cannot be described here in detail. A brief description of their principles/chapters/elements can be found in the Appendix A. It should be noted that, although many measurements within these protocols are similar, the interpretation and weight for the final scores differ substantially between the assessment protocols.

There are also welfare measures related to very specific parameters or behaviors. For instance, milk yield provides (indirect) information about cow welfare, and is correlated with body condition score [18,19]. Milk yield has also been found to be influenced by the posture of the cow [20]. Posture, i.e., lying or standing, influences the blood flow to the udder. An increased blood flow, during lying, stimulates milk production. Because lying time is influenced by the quality of the lying area, milk yield could be a reflection of a cow’s welfare. Cook et al. [21] defined the Stall Standing Index (SSI) as the percentage of cows standing in the freestalls. They proposed an impaired welfare if over 20% of the cows were standing when the SSI was measured up to two hours before milking. This percentage may be related to the lameness of cows on a farm, as lying activity is significantly impaired when a cow is lame [22]. The correlations between all these welfare assessment systems, whether being the end result of a protocol or a single measure, appeared to be very low [23].

Moberg [24] and Broom and Johnson [25] note that the effect of stress on animal welfare is difficult to interpret. Stress is usually seen as detrimental for the welfare of an animal, but it can also be good. Biological functions will change because of stress and can directly affect the animal’s welfare. The changes in biological function during stress results in a shift of biological resources and this change in biological function during stress is the ‘biological cost of stress’ [24]. For most stressors the biological cost is negligible because the stressors are short-lived. During prolonged stress or if stress is severe, the biological cost is substantial and the impact of stress becomes a significant burden to the body. A relatively brief exposure to a single stressor (acute stress) is usually considered as harmless, it can even be experienced as pleasant. The exposure to a series of acute stressors accounts for most chronic stress in animals [24]. It is also important whether the animal can easily adapt to the stressor or whether adaptation is associated with high costs [25]. The increase in cortisol becomes critical to the animal’s welfare only if the cortisol level is elevated for a sufficient time to cause a significant biological cost by shifting energy away from other biological functions. If this happens, the resulting distress affects animal welfare negatively [24].

Because stress elevates cortisol, the use of cortisol levels as index for cow welfare was investigated [26,27,28,29]. Cortisol is mostly measured in blood samples. However, in order to collect blood samples the animal needs to be handled, which could in itself be stressful and elevate the cortisol level in the blood [30,31]. Cortisol or the metabolites can also be determined with non-invasive sampling methods in saliva, faeces and urine [32]. However, measurements in blood samples or non-invasively, in saliva samples, reflect an acute activation of the Hypothalamo-pituitary-adrenal axis (HPA axis). Acute stress is not always bad for the welfare of an animal, it can enhance the welfare as well as mentioned above [24]. Chronic stress, however, is usually detrimental for the welfare of an animal [25]. For measuring cortisol levels over a prolonged period of time, a non-invasive sampling technique is needed. Cortisol concentration in hair samples has been proposed to reflect cortisol over a prolonged period of time [28,33,34]. Dairy cattle hair grows approximately 0.6 to 1 cm per month [35], but this varies among season and body location [36], with a complete molt every three months [37]. Therefore, a 2–4 cm hair specimen should reflect cortisol levels from a period of approximately three months. Raul et al. [38] were the first to describe detection of endogenously produced cortisol in human hair, while Koren et al. [39] described a different detection method for measuring cortisol in wildlife. Recently, this technique has also been used in dairy cows to evaluate levels of cortisol in the transition of winter housing to summer grazing [40] and has been standardized by Moya et al. [36]. It was found that hair cortisol was significantly higher in physiologically or clinically compromised cows, providing evidence for increased activation of the HPA axis upon (long lasting) stress in cows [28]. This was confirmed by Sharma et al. [41], who performed a study with (old) cows in Indian shelters, where the environmental conditions are sometimes far from optimal, and concluded that hair cortisol level could be a useful biomarker of stress in cows when conducting studies under field conditions.

The present study aimed to answer the question of whether hair cortisol and the outcomes of welfare assessment protocols described above, namely the original WQ, WQ-Mod, WW, WM, CWM, KK, CCSS, SSI, WI Tuyttens and EV are correlated, i.e., whether hair cortisol measures can be used as a valid physiological index of cow welfare. In addition, we determined the correlations of results of these welfare assessment protocols.

These welfare assessment protocols and corresponding hair cortisol concentrations were collected from a wide range of farms. Based on previous reports [27,28,29,33,42], it is hypothesized that the result of the welfare protocols will be negatively correlated with hair cortisol because most welfare assessment protocols investigated here yield a higher score when the welfare level of the cows is higher. The only exception is the SSI which is hypothesized to correlate positively with hair cortisol, because an impaired welfare is proposed if over 20% of the cows are standing [21]. Some welfare assessment protocols base their final score on the scores for main principles, such as feeding, housing, health and behaviour (WQ, WQ-Mod, WM and KK). For these assessment protocols, correlations between main principles/chapters and cortisol level were examined as well.

## 2. Materials and Methods 

### 2.1. Farm Selection

Four large veterinary practices spread out over The Netherlands participated in this study. Each of these practices had more than five veterinarians for the dairy farms. All practices created a list of farms, with Holstein Friesian cows, suitable for participation by excluding farms that would stop within 5 years, were refurbishing, had executed the KK protocol before or could not provide reliable information about their milk production. The farms on the list were randomized for the order in which to approach farmers for participation. All the dairy cattle veterinarians of the veterinary practices then classified these farms as weak (1 point), sufficient (2 points), or good (3 points), based on the following criteria: chronic undernourishment, poor housing, weak health and negative impression of the behaviour (such as a high level of aggression or fear). The classification was made by consensus of all the veterinarians (*n* > 5) that visit the dairy farms on a regular basis, but are not based on a formal assessment protocol. This estimation of the veterinarians (EV) is also used in the analysis. All veterinarians had background and practical knowledge of animal welfare. The first five farms on the list within each class were asked to participate. If a farmer refused, the sixth, seventh or eighth farm was then approached. Refusal only occurred for a few farms that had already been classified as weak by the vets. This resulted in 20 weak, 20 sufficient and 20 good farms in total, the average number of cows per farm was 86 (range 27–224) [11]. The farms did not have groups for production level or parity.

### 2.2. Executing Welfare Assessment Protocols

For execution of the WQ, KK and WW protocols seven observers (veterinarians) were trained (WQ by the Welfare Quality consortium, KK by the KK consortium, WW has overlapping parameters with both protocols). The observers were recruited from the participating veterinary practices but observed the farms of another practice. These observers monitored animal welfare according to the protocols of WQ, WQ-Mod, KK, WW, CCSS and SSI. One farm was visited by a different observer for each protocol (WQ, WW & KK). CCSS and SSI were calculated out of the results of the other protocols. The data for the CWM were obtained from the external databases and WI Tuyttens was calculated on the basis of the WQ data. For the animal-based observations, a percentage, based on farm size, was randomly selected from all cows according to WQ. The raw data were processed, and an end result was calculated per welfare assessment protocol per farm, and WM was computed.

### 2.3. Collecting Hair Samples

For collection of the hair samples, the observers were instructed to shave a square of 2 cm × 2 cm on the flank of the cow (Figure 1). The square had to be clean and the hairs had to be white because, according to Bennet and Hayssen [43], lighter hairs provide a higher and thus more easily detectable cortisol concentration. All samples were kept at room temperature in the dark until processed. Samples were collected from the first ten cows that were selected for the animal-based observations (as described in the WQ protocol) and fulfilled the inclusion criteria (clean and white hair within the square of Figure 1). Heifers were excluded because it was not ascertained that they had been in the same environment for the past half year. No data were collected about age, parity, milk yield, and days in milk. All samples were collected in the period April–May 2014. Two farms dropped out of the project and some samples were missing or not workable, resulting in 548 individual hair samples.

Due to time restraints and the workload of processing the samples, five samples per farm were measured within four months after sampling, while the other five samples per farm were measured approximately one year after collection of the samples. The samples were stored in a dry and dark place at room temperature without any (pre)processing. To determine whether storing time of the samples affected the cortisol concentrations measured, cortisol concentrations of fifteen randomly selected samples were analyzed during both runs.

### 2.4. The (Pre)Processing of the Hair

Before quantification of cortisol in hairs, the samples had to be processed extensively first. In short, approximately 250 mg of hair was placed in a 50 mL Cellstar tube (Greiner Bio-one, Kremsmünster, Austria) covered with aluminium foil for protection against UV-light. After washing with 10 mL 2-propanol (1.09h634, Emsure®, Merck, Kenilworth, NY, USA) the hairs were dried at 37 °C for four days. Next, approximately 35 mg of clean hair was weighed off and placed in a microtube (clickcap, 2.0 mL Trefflab, Nolato Treff AG, Degersheim, Switserland). Three beads (3.2 mm, Cat. No 11079132 BioSpec Products, Bartlesville, OK, USA) were added to each tube. The hairs were now ground with a Tissuelyser II (Cat. No 85300, Qiagen, Hilden, Germany). For cortisol extraction [43] the hair samples were incubated with slow rotation for 24 h with 1 mL of methanol (1.06009.2500, Emsure). After centrifuging of the samples, 600 μL of the supernatant were transferred to a new tube. The extracts were dried in a Speedvac (Automatic Environmental Speedvac, AES1000, Savant Instruments, Woonsocket, RI, USA) with medium temperature for 2.5 to 3 h. The precipitate was then dissolved in 100 μL of phosphate buffer (assay diluent provided in the assay kit) [36] for 24 h on a shaker set at 300 rpm (KS250, Janke & Kunkel, Breisgau, Germany).

### 2.5. Analyzing Cortisol Concentrations

Hair samples were analyzed according to a protocol that was first described by Koren et al. [39] and later by Davenport et al. [44]. This entails the use of a high sensitivity salivary cortisol EIA kit (Salimetrics, Carlsbad, CA, USA). The intra-assay CV was 10.5% (low), 6.6% (medium) and 7.3% (high).

Samples from all farms were measured in duplicate and evenly distributed over the plates. Non-specific binding (NSB) wells were included for each plate and a high and low control provided by the kit were run on every plate. According to the protocol provided by Salimetrics, 25 µL of standards, unknowns and control were pipetted in the appropriate wells. Also 25 µL of assay diluent was pipetted into the NSB wells and the zero wells. Then 200 µL of enzyme conjugate dissolved in assay diluent was added to each well. This was mixed for 5 min on a plate shaker at 500 rpm (Microplate shaker, PMS-1000i Grant bio, Royston, UK) after which it was incubated for an additional 55 min at room temperature. This allowed the cortisol in the samples and the conjugate to bind with the antibodies coated on the well. This means that if there were high levels of cortisol in the sample, little conjugate could bind and if there were low levels of cortisol, high levels of conjugate could bind [45]. Wells were washed with a wash buffer to remove all unbound cortisol and antigen. Then, 200 µL of 3,3′,5,5′-tetramethylbenzidine (TMB) was added to each well, mixed on the plate shaker for 5 min at 500 rpm (covered with aluminium foil), and incubated for 25 min in the dark. Then, 50 µL of 0.2 M H_2_SO_4_ was added to each well to stop the enzymatic reaction and mixed on the plate shaker set at 450 rpm for 3 min. Optical density was measured at 450 nm and background optical density was measured at 620 nm with a multimode detector plate reader (DTX880, Beckman Coulter, Brea, CA, USA).

### 2.6. Calculating Cortisol Concentrations

The optical density that was measured with the assay kits had to be converted to an actual cortisol concentration. To begin with, the background optical density (620 nm) (BO) was subtracted from the optical density for determining cortisol concentrations (450 nm). After that, the average optical density was computed for each sample. The average optical density from the non-specific binding (NSB) wells was then subtracted from each average optical density to account for any arbitrary binding to the wells. Subtracting the NSB from the zero wells gave rise to the B0. Subtracting the NSB from all other wells gave rise to the B corresponding to each sample, standard and control. Now, the ratio between the average optical density of cortisol and the zero binding (B/B0) could be calculated for each sample, standard and control. For each of the standards, the logarithm of the concentration was taken. Taking this logarithm as the x-axis and the B/B0 of the standards for the y-axis, a calibration graph was computed. From the computed B/B0s for each unknown and control, the corresponding logarithm of the concentration was interpolated with GraphPad Prism (version 2.6 San Diego, CA, USA). This value was then used to compute the concentration in pg/mL. For the unknowns, the concentration was corrected with a factor of 1/6 for taking 600 µL of the supernatant and for dissolving the precipitate in 100 µL of assay diluent. Lastly, the concentration was divided by the weight of the hairs from which the cortisol was extracted in the first place. This generated the concentration of cortisol in pg per mg of hair.

### 2.7. Statistical Analysis

All statistical analyses were performed using SAS 9.4 (SAS Institute Inc., Cary, NC, USA) running on a Windows 10 platform. A repeated measures ANOVA with the factor “Storage time” was computed on the cortisol values of the 15 samples that were measured at both timepoints, to determine whether there was a loss of cortisol during storage (using SAS PROC GLM).

#### Correlations between Analyzing the First and Second Series of Cortisol Concentrations

Mean cortisol concentrations per farm were calculated from the hair cortisol concentrations of the individual cows. Because the samples were processed and measured at 2 different time points after collection, the correlation between samples at timepoint 1 and timepoint 2 was determined with the Spearman rank correlation. In addition, the intra class correlation (ICC) was determined (SAS PROC MIXED). ICC is a measure of reliability that reflects both, the degree of correlation and agreement between the two measurements [46].

Spearman rank correlations (Rho, ρ) between mean cortisol per farm and summed welfare assessment scores (i.e., end scores) and scores of principles/sub-scores, (most of which are measured on the ordinal level), derived from the different welfare assessment tools were calculated (SAS PROC CORR). The correlations were subdivided into four different classes: (1) correlations between hair cortisol and end results and selected sub-scores of different welfare assessments; (2) correlations between the welfare assessment system WQ and its derivatives WQ Mod, WI Tuyttens, WM; (3) correlations of WQ (derived) assessment systems with other welfare assessment scores, and (4) correlations between the all welfare assessments except the WQ and WQ derived systems. Note that the correlations in the first class explicitly address our hypotheses that hair cortisol measures can be used as a valid physiological index of cow welfare.

In addition, per welfare assessment tool, rank correlations between the cortisol measurements and selected (sub) chapters were calculated. Note that the correlations in classes 2 to 4 and the correlations between selected sub chapters were considered as exploratory and not as an explicit test of our hypothesis.

Rank correlation coefficients (ρ) in the range of −0.2 to −0.3, or of +0.2 to +0.3 were considered as poor; ρ between −0.4 and −0.5, or between +0.4 and +0.5 were considered as fair; ρ between −0.6 and −0.7, or between +0.6 and +0.7 were rated as strong; whereas ρ exceeding −0.8 or +0.8 were considered as very strong. A ρ of −1 or +1 reflects a perfect relationship between measures [47].

For the correlations in the first class, which are considered as direct test of our hypothesis, we applied a Bonferroni adjustment as well as the less conservative False Discovery Rate (FDR) adjustment to of the *p*-values to account for testing multiple (10) correlations.

## 3. Results

Hair cortisol concentrations for the entire experiment ranged from 3.93–127.42 pg/mg hair, mean and standard deviation were respectively 20.50 pg/mg and 0.775 for n = 548 cows. The average cortisol concentrations per farm ranged from 7.06–60.37 pg/mg, with a mean and standard deviation of 20.41 pg/mg and 1.923, respectively, for n = 58 farms.

### 3.1. Effect of Storage Time of 15 Hair Samples on Cortisol Concentrations

Samples measured in the second run, 12 months after the first run, had significantly lower cortisol levels compared to samples measured at the first time point (F1,14 = 7.92; *p* = 0.0138; Figure 2, panel A). This indicates a decrease of cortisol concentration in hair samples due to storage time.

Despite the loss in signal, there was a strong rank correlation (ρ = 0.778) and an intra class correlation (ICC) of 0.73, indicating moderate to good reliability [46] between the mean cortisol values from the first five samples and the second five samples across the 58 farms (Figure 2, panel B).

### 3.2. Correlations between Hair Cortisol and the End Results/End Scores of Different Welfare Assessment Protocols

Most rank correlations between hair cortisol and the scores yielded small negative values, i.e., with an associated probability > 0.05 (WI Tuyttens, WW, CWM, KK, CCSS, SSI; all with *p* > 0.05; see Table 2, dark yellow areas). In weak support of or hypothesis that the result of the welfare protocols are negatively correlated with hair cortisol because most welfare assessment protocols investigated here yield a higher score when the welfare level of the cows is higher, the strongest correlation between cortisol and a welfare index was with the estimate of the vets (ρ = −0.284; *p* < 0.031). However, after correction of *p*-values for multiple correlations, this correlation was no longer significant. Moreover, contrary to expectations, a fair positive rank correlation was found between WQ-Mod, ρ = 0.434; *p* < 0.001, and the WM end result, ρ = 0.455; *p* < 0.001 and hair cortisol. Only negative correlations would have supported our hypothesis.

### 3.3. Corrleations between Hair Cortisol and Sub-Scores and Principles of Various Welfare Assessment Protocols

Of the correlations between hair cortisol and principles of WQ and WQ-Mod, subscores of WW and KK, scores of CWM and chapters of CCSS, only a few reached statistical significance (see Table 3). The pattern of correlations between the cows’ hair cortisol and the principles in WQ and their corresponding principles in WQ Mod was highly similar. Hair cortisol levels had a poor negative correlation with the WQ Mod principle health, ρ = −0.264, *p* ≤ 0.045, while the correlation of hair cortisol with the WQ principle health failed to reach statistical significance, ρ = −0.231, *p* ≤ 0.081. The principle behaviour of both WQ and WQ Mod had a fair positive correlation with hair cortisol (WQ: ρ = −0.474, *p* ≤ 0.001; WQ Mod: ρ = −0.442, *p* ≤ 0.001). Whereas hair cortisol showed a poor negative correlation with the WW subscore for mastitis, ρ = −0.284, *p* ≤ 0.031, no relationship was found with the corresponding KK subscore, ρ = −0.093, *p* ≤ 0.489. The WW production subscores increased with decreasing hair cortisol levels, ρ = −0.357, *p* ≤ 0.006; a finding that was in line with the fair negative correlation between the CWM score Economic result and hair cortisol levels, ρ = −0.431, *p* ≤ 0.001.

### 3.4. Correlations between Cortisol and Freestall-Related Measures, and Several Health-Related Parameters at Herd Level

None of the freestall-related measures correlated with the cows’ hair cortisol levels (see Table 4A). Also, most health-related measures of the different welfare assessments systems had no relatonship with hair cortisol (see Table 4B), except for the WQ Mod principle health that had a poor negative correlation with hair cortisol (as already detailed in the previous paragraph).

A striking finding was that the correlations between cortisol and access to pasture were positive in both the KK and WQ protocols (KK: ρ = 0.348, *p* < 0.007; WQ: ρ = 0.646; *p* < 0.001, resp; see Table 4C).

### 3.5. Correlations between the Results of the Welfare Assessment Protocols

#### 3.5.1. Correlations between the Welfare Assessment System WQ and Its Derivatives, Namely WQ Mod, WI Tuyttens and WM

One might expect that the end results of the welfare assessment using these systems are correlated (see Table 2, light yellow area). Except the end result of WI Tuyttens that did not correlate with any of the other WQ (derived) assessment systems, we found the expected positive correlations with WQ, ranging from fair (WQ—WQ Mod: ρ = 0.413, *p* ≤ 0.001; WQ—WM: ρ = 0.484, *p* ≤ 0.001) to very strong (WM—WQ Mod, ρ = 0.976, *p* ≤ 0.001).

#### 3.5.2. Correlations of WQ (derived) Assessment Systems with Other Welfare Systems and Scores

WQ Mod had a poor negative correlation with the KK subscore welfare, ρ = −0.303, *p* ≤ 0.021 (Table 2, light gray area). WI Tuyttens, however, showed a fair to strong correlation with WW: ρ = 0.569, *p* ≤ 0.001; CWM: ρ = 0.481, *p* ≤ 0.001; KK subscore welfare: ρ = 0.638, *p* ≤ 0.001; CCSS: ρ = 0.562, *p* ≤ 0.001; and EV: ρ = 0.587, *p* ≤ 0.001).

#### 3.5.3. Correlations between the Welfare Assessment Systems except WQ and WQ Derived Systems 

WM had a poor negative correlation with the KK subscore welfare, ρ = −0.266, *p* ≤ 0.043 (Table 2, dark gray area). WW had a poor positive correlation with CWM, ρ = 0.337, *p* ≤ 0.01; a fair positive correlation with the KK subscore welfare, ρ = 0.497, *p* ≤ 0.001, a fair positive correlation with CCSS, ρ = 0.399, P ≤ 0.002, and strong positive correlation with EV, ρ = 0.692, *p* ≤ 0.001. 

CWM also correlated poorly with WW, ρ = −0.337, *p* ≤ 0.01, KK, ρ = 0.327, *p* ≤ 0.012, CCSS, ρ = 0.294, *p* ≤ 0.025, and fairly positive with EV, ρ = 0.449, *p* ≤ 0.001.

The KK subscore welfare showed a poor negative correlation with WM, ρ = −0.266, *p* ≤ 0.043, a fair positive correlation with WW, ρ = 0.497, *p* ≤ 0.001, a poor positive correlations with CWM**,** ρ = 0.327, *p* ≤ 0.012, a strong positive correlation with CCSS**,** ρ = 0.662, *p* ≤ 0.001, a fair positive correlation with SSI, ρ = 0.433, *p* ≤ 0.001 and a fair positive correlation with EV, ρ = 0.441, *p* ≤ 0.001

CCSS showed a fair positive correlation with WW, ρ = 0.399, *p* ≤ 0.002, a poor positive correlation with CWM, ρ = 0.294, *p* ≤ 0.025, a strong correlation with the KK subscore welfare, ρ = 0.662, *p* ≤ 0.001, a fair correlation with SSI, ρ = 0.576, *p* ≤ 0.001, and a fair correlation with the EV, ρ = 0.448, *p* ≤ 0.001.

SSI (standing idle) had a fair positive correlation with the KK subscore welfare, ρ = 0.433, *p* ≤ 0.001, and with CCSS, ρ = 0.576, *p* ≤ 0.001.

EV correlated strong and positive with WW, ρ = 0.692, *p* ≤ 0.001, fair with CWM, ρ = 0.449, *p* ≤ 0.001, CCSS, ρ = 0.441, *p* ≤ 0.001, and with SSI, ρ = 0.448, *p* ≤ 0.001.

## 4. Discussion

Measuring hair cortisol concentrations in cattle has been performed several times [28,33,36,40,48]. The aim of the present study was to determine whether hair cortisol concentrations may be taken as a physiological measure of cow welfare. This was done by determining the correlations between mean hair cortisol concentrations and various existing farm welfare assessment protocols. Some authors [28] associated increased levels of cortisol with stress and consequently expect a negative correlation between hair cortisol levels and good welfare. This notion is supported by a study of González de la Vara et al. [49] who showed that ACTH stimulation in cattle increases hair cortisol levels. Therefore, hair cortisol could be used as an indicator for HPA axis activation during stressful circumstances. As mentioned before, a 2–4 cm hair specimen should reflect cortisol levels from approximately a three month-period. 

At each farm 10 hair samples were collected and stored in sealed plastic bags in the dark at room temperature. The area used for the sampling was as indicated in Figure 1. It was impossible to get a sample of the exact same spot, because the location of black and white hair varies among individuals. In this way, a sample from approximately the same part of the skin was collected. The thickness of the skin is similar throughout this area. By taking a sample of the first ten suitable cows (clean white hair within the square of Figure 1 and parity > 1), a random sample of the herd was used. This means that there was no selection for other parameters and we assume that the average number of days in milk would be similar for all the farms, because all had a year round calving pattern. Five hair samples per farm were measured immediately. The other samples were analyzed after one year. The cortisol values of the first five samples were significantly higher than the cortisol values of the second five samples, indicating a loss of cortisol over time. This result is in contrast to findings by González de la Vara et al., who didn’t observe difference in cortisol concentrations between hair samples stored for 30 days at room temperature and hair samples stored for 12 months [49]. Others, however, also found an effect of storage time [50,51]. Abell et al. [50] stored their samples for more than 18 or 24 months before assay. It is not mentioned how they were stored. Samples were stored in paper envelopes in a dark dry location until the date of extraction in the study of Azevedo et al. [51]

Welfare assessment scores are not as objective as hair cortisol concentrations. Although many efforts have been made to create valid and reliable protocols to measure welfare, results of these protocols are still subject to debate [7,52,53]. This is demonstrated by the very low correlations between the outcomes of most of the various protocols, corroborating earlier findings [17].

Although we expected negative correlations between mean cortisol concentrations and the welfare assessment scores, the results showed only two significant correlations, of which the highest was a positive correlation for WQ-Mod end result (ρ = 0.43; *p* < 0.001). The only significant negative correlation was with the estimate of the vets (ρ = −0.28; *p* < 0.05). However, this correlation failed to reach significance if appropriate corrections for multiple correlations were applied. Nevertheless, this correlation suggests that veterinarians were able to estimate the level of stress at the farms. It was not the intention of this study to compare the opinion of the vets with the other protocols or cortisol, but it was needed and used to select the farms in order to obtain variability in the level of welfare/stress at the farms included in this study.

The welfare component of KK had a trending negative correlation (ρ = −0.23; *p* < 0.085) with the hair cortisol level. This is an aggregation of several parameters: activity, BCS, locomotion, swollen hocks, hygiene, deviating cows, and general impression [12]. There was a fair correlation between the welfare component of KK and the EV (ρ = 0.44; *p* < 0.001). The EV was based on some of these parameters.

Milk yield and economic results (included in CWM) of the farm showed a negative correlation with cortisol. Which is confirmed by Fukasawa et al. [54], who reported a negative correlation between milk protein content and cortisol (in milk). Caroprese et al. [55] studied this correlation in sheep, which also showed a negative correlation between cortisol and milk yield. This relationship should motivate farmers to reduce stress, i.e., improve the welfare level of the animals, and increase earnings [56].

The low correlations we found between health parameters and cortisol was not as expected [33]. That the correlations with the health-related parameters (Table 3C) were inconsistent and not statistically significant is remarkable since health is one of the key components of welfare. Burnett et al. [33] reported that multiparous cows had higher cortisol levels in hair than primiparous cows. In our selection of the cows we did not include parity (primiparous cows excluded) nor the number of days in milk (DIM). However, these have a minimal influence on hormone parameters [57]. Burnett et al. [33] reported an effect of DIM and pregnancy for primiparous cows, but the effect in multiparous cows was minimal. Furthermore, they measured cortisol in 3 week periods. Glucocorticoids from the HPA axis rise due to immune cell activation and due to production of pro-inflammatory cytokines by damaged tissues during disease [58,59]. Other studies described that inflammatory mediators induced by mastitis, increase the release of CRH and thereby activate the HPA-axis [60,61]. Our results are not in concordance with a study of Comin et al. [31] who found a significant negative correlation between clinical mastitis and hair cortisol in cows. A clinical case of mastitis usually lasts for a few days and will thus not increase cortisol levels over a long period of time. Subclinical mastitis, as reflected in the somatic cell count, may persist for a long period of time and can thus be related to higher cortisol levels. However, Burnett et al. did not find a relation between subclinical problems and cortisol in hair [33]. Furthermore, in the study of Comin et al. [31], other diseases were included as well. All cows were scored for locomotion, but if they were lame we do not know for how long this situation had existed. If the cow became lame recently, or healed recently, this will not be reflected in the cortisol levels in the hair. One has to consider, however, that the health parameters are measured on a large number of individual cows and the hair samples were also taken from individual cows. The number of ten cows for the hair samples might have been too low to get a significant correlation. This is certainly not true for the correlations with the freestalls (Table 3B), because all cows have to deal with the same freestalls. The correlations with the freestall components and the overall assessment in the CCSS were low and nonsignificant, except for the softness of the bedding (ρ = −0.25; *p* = 0.05). This is an important feature of a freestall and substantially influences the time spend lying [62,63,64,65]. 

It was a striking finding that cortisol correlated positively with access to pasture. This was not expected based on the assumption that cows appreciated access to pasture [66]. However, the hair samples were taken at the end of the barn period. The farms that give the cows no access to pasture may be the ones with the best indoor conditions explicitly because that is the only housing area for the animals.

We found just one (KK-welfare) negative correlation between cortisol and the end result of the welfare assessment protocols. One of the main potential issues for interpretation of these correlations is the difference between stress and welfare. Cortisol is a stress indicator, but welfare is not solely based on the absence of stress [29,32]. Cortisol is affected by social rank, especially for high-ranking cows [67]. It is also affected by pregnancy state and lactation state [33,40,49]. Serum cortisol values are significantly higher in the late stage of pregnancy [33,40,49], which affects the cortisol concentrations until 90 days postpartum [54]. Because we randomly took hair samples from ten cows and did not take social rank, lactation state and pregnancy state in consideration, this could explain some of the extreme high hair cortisol values found. For further investigation, hair samples should be collected from non-high-ranking cows more than 90 days in milk. 

Mean hair cortisol concentrations in cows vary between published studies, but they all range between 0.3–20.4 pg/mg hair [28,33,36,40,48], which is low compared to the cortisol concentrations found in the present study (3.93–127.42 pg/mg hair). This could be due to a different sample preparation. For example, Tallo Parra et al. [48] minced the hair into <2 mm fragments; in the present study hair were ground with a ball mill, thereby expanding the surface area of the hair, which may allow more access to cortisol for measurement [33].

We found a wide range of cortisol values between farms and even within farms. This is in concordance with a study were a wide range within one herd was found [28]. A possible explanation for this are individual differences in HPA axis response. Individual difference in HPA axis response has been well described in humans, rats, mice and different farm animals [29,68]. Stress and HPA axis activation has been extensively studied, but when it comes to chronic stress, the results are very inconsistent [26,29]. Some studies report increased HPA axis response during chronic stress [26,44], while others describe a blunted HPA axis response [26,29]. Further investigation is needed to determine how exposing cattle to chronic stress affects (hair) cortisol concentrations [69].

## 5. Conclusions

In conclusion, our hypthesis that hair cortisol might be taken as a valid measure of cow welfare was not supported. Because almost none of the welfare assessment protocols or chapters correlated negatively with cortisol, they seem not to rely on measures related to stress, an important component of welfare. Furthermore, the welfare assessment protocols did not correlate well with each other, corroborating earlier findings [17]. This is remarkable, since all these assessment tools claim to provide measures for animal welfare, whereas they apparently measure something else. Consequently, further research is needed for identifying the conditions, under which hair cortisol may be taken as measure of stress in cows, i.e., to evaluate how hair cortisol levels are influenced by chronic stress in cattle. Unfortunately, the available welfare assessment protocols might not provide reliable tools for measuring welfare and should be re-evaluated, eventually modified and thoroughly validated.

## Figures and Tables

**Figure 1 animals-11-00821-f001:**
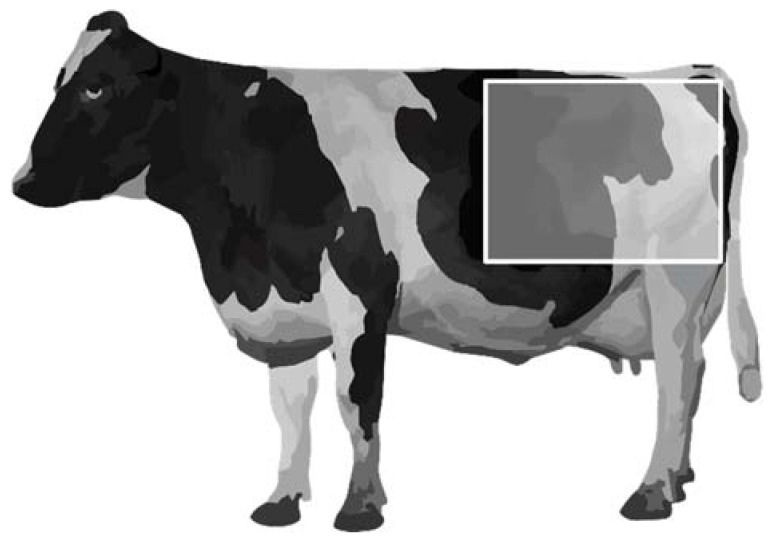
Site from which the hair samples were derived. Only clean, white hairs were collected.

**Figure 2 animals-11-00821-f002:**
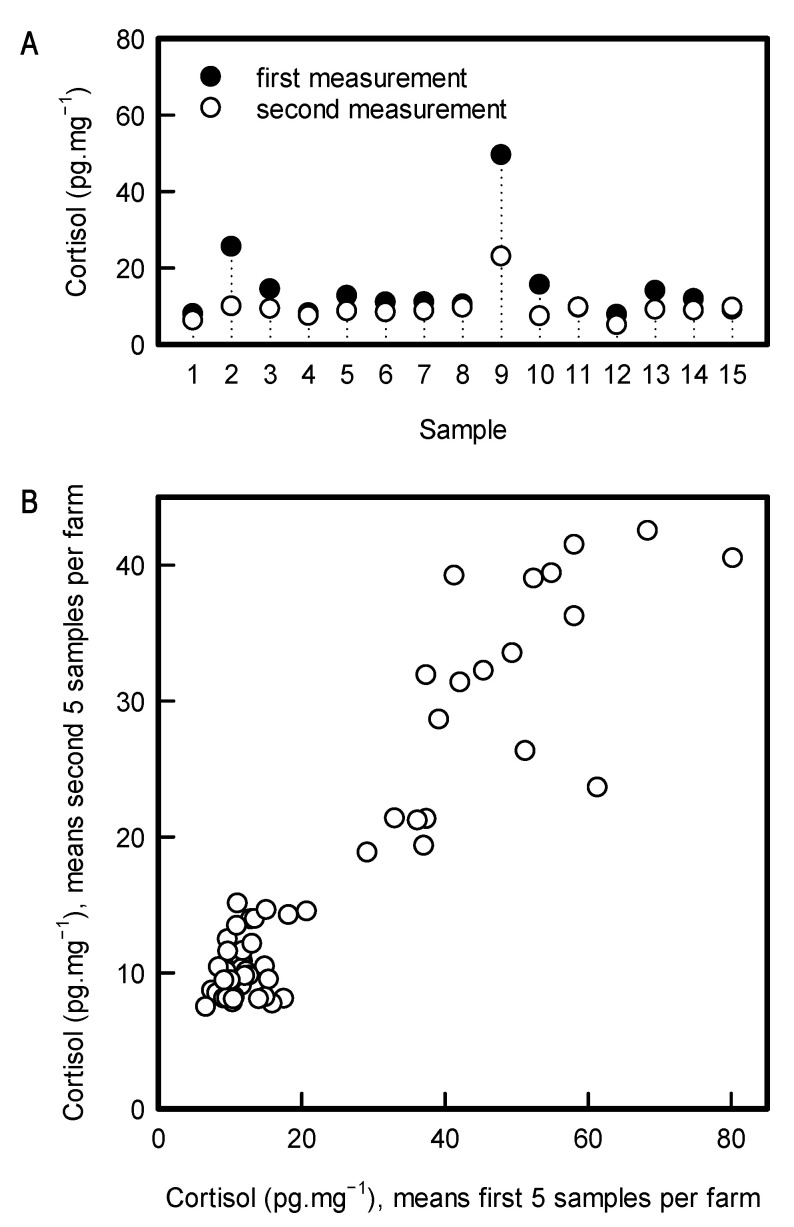
Panel (**A**): Difference in cortisol values from the 15 samples processed and measured within 4 months and one year after sampling. Panel (**B**): Pairwise plot of the untransformed hair cortisol means of the first 5 samples and means of hair cortisol values of second 5 samples per farm.

**Table 1 animals-11-00821-t001:** The welfare assessment systems used in this study.

Abbreviation	Name of Protocol	Data Used	Refs.
WQ	Welfare Quality	Obtained on farm	
WQ Mod	Modified version of WQ	Similar to WQ	[11]
WI Tuyttens	Welfare Index	Based on WQ	[16]
WM	Welfare Monitor	Obtained on farm	[17]
WW	Welfare Indicator	Obtained on farm	*
CWM	Continuous Welfare Monitor	Various data bases	[7]
KK	Cow Compass	Obtained on farm	[8,12]
CCSS	Cow Comfort Scoring System	Obtained on farm	[15]
SSI	Stall Standing Index	Obtained on farm	**
EV	Estimate Vets	Based on previous experience	

*: https://www.wur.nl/nl/show/Welzijnswijzer-Melkvee.htm (accessed on 12 March 2021); **: Also named Cow Comfort Index (CCI) https://www.progressivedairy.com/topics/barns-equipment/using-indices-to-assess-freestall-comfort (accessed on 12 March 2021).

**Table 2 animals-11-00821-t002:** Spearman rank correlations (Rho, ρ) and associated probabilities between hair cortisol concentrations, relevant parameters of the various welfare assessment protocols (end results) and the estimates of the veterinarians, from 58 different farms. Correlations with associated probability ≤ 0.05 are printed bold. The only correlation supporting the hypothesized negative relationship between hair cortisol levels and a welfare score is printed bold.

		Average Cortisol	WQ: End Result	WQ Mod: End Result	WI Tuyt-tens	WM: End Score	WW: End Score	CWM: End Result	KK: Sub-Score Welfare	CCSS: Total	SSI: Stand-ing idle	EV: Estimate Vets
**Average cortisol**	ρ	1.000	0.094	**0.434**	−0.167	**0.455**	−0.175	−0.132	−0.229	−0.016	−0.150	**−0.284**
	*p*≤		0.482	0.001	0.209	0.001	0.190	0.325	0.084	0.906	0.260	0.031
**WQ: End result**	ρ	0.094	1.000	**0.413**	0.209	**0.484**	0.190	0.098	−0.010	0.067	−0.051	0.139
	*p*≤	0.482		0.001	0.115	0.001	0.153	0.463	0.941	0.616	0.702	0.298
**WQ: Mod: End result**	ρ	**0.434**	**0.413**	1.000	−0.154	**0.976**	−0.019	−0.192	**−0.303**	−0.083	−0.096	−0.027
	*p*≤	0.001	0.001		0.250	0.001	0.887	0.149	0.021	0.536	0.472	0.842
**WI Tuyttens**	ρ	−0.167	0.209	−0.154	1.000	−0.122	**0.569**	**0.481**	**0.638**	**0.562**	0.209	**0.587**
	*p*≤	0.210	0.115	0.250		0.360	0.001	0.001	0.001	0.001	0.115	0.001
**WM: End score**	ρ	**0.455**	**0.484**	**0.976**	−0.122	1.000	−0.037	−0.214	**−0.266**	−0.050	−0.078	−0.070
	*p*≤	0.001	0.001	0.001	0.360		0.781	0.107	0.043	0.709	0.560	0.599
**WW: End score**	ρ	−0.175	0.190	−0.019	**0.569**	−0.037	1.000	**0.337**	**0.497**	**0.399**	0.085	**0.692**
	*p*≤	0.190	0.153	0.887	0.001	0.781		0.010	0.001	0.002	0.524	0.001
**CWM: End result**	ρ	−0.132	0.098	−0.192	**0.481**	−0.214	**0.337**	1.000	**0.327**	**0.294**	0.144	**0.449**
	*p*≤	0.325	0.463	0.149	0.001	0.107	0.010		0.012	0.025	0.282	0.001
**KK: Sub-score welfare**	ρ	−0.229	−0.010	**−0.303**	**0.638**	**−0.266**	**0.497**	**0.327**	1.000	**0.662**	**0.433**	**0.441**
	*p*≤	0.084	0.941	0.021	0.001	0.043	0.001	0.012		0.001	0.001	0.001
**CCSS: Total**	ρ	−0.016	0.067	−0.083	**0.562**	−0.050	**0.399**	**0.294**	**0.662**	1.000	**0.576**	**0.448**
	*p*≤	0.906	0.616	0.536	0.001	0.709	0.002	0.025	0.001		0.001	0.001
**SSI: Standing idle**	ρ	−0.150	−0.051	−0.096	0.209	−0.078	0.085	0.144	**0.433**	**0.576**	1.000	0.127
	*p*≤	0.260	0.702	0.472	0.115	0.560	0.524	0.282	0.001	0.001		0.344
**EV: Estimate vets**	ρ	**−0.284**	0.139	−0.027	**0.587**	−0.070	**0.692**	**0.449**	**0.441**	**0.448**	0.127	1.000
	*p*≤	0.031	0.298	0.842	0.001	0.599	0.001	0.001	0.001	0.001	0.344	
			Correlations between hair cortisol and end results and selected sub-scores of different welfare assessments									
			Correlations between the welfare assessment system WQ and its derivatives WQ Mod, WI Tuyttens, WM									
			Correlations of WQ (derived) assessment systems with other welfare assessment scores									
			Correlations between the all welfare assessments except the WQ and WQ derived systems									

Note that after adjusting the *p*-values of the correlations between hair cortisol and the different welfare assessment systems using the Bonferroni adjustment as well as the less conservative False Discovery Rate (FDR) adjustment, the correlation between hair cortisol and the estimate of the veterinarians was no longer significant (Benjamini-Hochberg adjusted *p*-value: 0.103).

**Table 3 animals-11-00821-t003:** Spearman rank correlations (Rho, ρ) and associated probabilities (*p*-values, printed bold if *p* ≤ 0.05) between cortisol and (A) WQ principles, (B) WQ-Mod principles, (C) WW subscores, (D) KK subscores, (E) CWM scores and (F) CCSS chapters (all variables at herd level).

**(A) WQ Principles**	**ρ**	***p*** **≤**
Feeding	0.103	0.443
Housing	−0.155	0.247
Health	−0.231	0.081
Behaviour	**0.474**	**0.001**
**(B) WQ-Mod principles**	**ρ**	***p*** **≤**
Feeding	0.152	0.256
Housing	−0.155	0.247
Health	**−0.264**	**0.045**
Behaviour	**0.442**	**0.001**
**(C) WW subscores**	**ρ**	***p*** **≤**
Production	**−0.357**	**0.006**
Body Condition Score	−0.057	0.672
Mastitis	**−0.284**	**0.031**
Lameness	−0.137	0.304
**(D) KK subscores**	**ρ**	***p*** **≤**
Mastitis	−0.093	0.489
Cell count	0.000	0.999
Housing youngstock	−0.240	0.075
Locomotion score	−0.129	0.333
Welfare	−0.229	0.084
Housing	0.083	0.535
**(E) CWM scores**	**ρ**	***p*** **≤**
Tank Milk Cellcount	0.129	0.336
Econ result	**−0.431**	**0.001**
Deaths	−0.035	0.792
Non return	0.168	0.217
Calving Interval	−0.063	0.648
**(F) CCSS chapters**	**ρ**	***p*** **≤**
General	−0.145	0.276
Lameness	−0.086	0.522
Health & feeding	0.063	0.638

**Table 4 animals-11-00821-t004:** Spearman rank correlations (Rho, ρ) and associated probabilities (*p*-values, printed bold if ≤ 0.05) between cortisol and (A) freestall-related measures, (B) several health-related parameters, taken from the different welfare assessment protocols, and (C) access to pasture, all at herd level. The correlations supporting the hypothesized negative relationship between hair cortisol levels and a welfare score are printed bold and red.

**(A) Freestall-Related Measures**	**ρ**	***p*** **≤**
Number freestalls (WW)	−0.023	0.849
Freestall length (WW)	−0.042	0.754
Freestall width (WW)	0.077	0.563
Freestall diagonal (WW)	−0.063	0.640
Bedding softness (WW)	−0.254	0.054
Deep litter (WW)	0.129	0.335
Freestalls (CCSS)	−0.041	0.759
**(B) Health-related measures**	**ρ**	***p*** **≤**
Health (WQ Mod)	**−0.264**	**0.045**
End score health (KK)	−0.060	0.660
Disease incidence (KK)	−0.113	0.406
Mortality (WQ)	−0.168	0.209
Deaths (CWM)	−0.036	0.792
Disease death (WW)	−0.166	0.214
Culling (KK)	0.191	0.150
Severely lame (WQ)	0.210	0.114
Not lame (WQ)	−0.073	0.585
Lameness (CCSS)	−0.086	0.522
Locomotion score (KK)	−0.151	0.259
Lameness (WW)	−0.137	0.304
Tank Milk Cell count (CWM)	0.129	0.336
**(C) Pasture-related measures**	**ρ**	***p*** **≤**
Pasture (KK)	**0.348**	**0.007**
Access to pasture (WQ)	**0.646**	**0.001**

## Data Availability

Not applicable.

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
