# Peer review of "The Relation between Hair-Cortisol Concentration and Various Welfare Assessments of Dutch Dairy Farms"

_animals, 2021, doi:10.3390/ani11030821_

Round 1

Reviewer 1 Report

The authors have improved this manuscript and added some very important details, but I still believe it is missing information to make it transparent and thus repeatable for other researchers. Overall it is still quite well written although the introduction could be more concise. There are many limitations to this study. These limitations should be clear to the reader (e.g. no control was taken for which cows were sampled interms of lactation status, health status, percentage of the cows sampled, welfare assessments are very broad etc).

As mentioned in the last revision (and also by other reviewers as well), I think you need to state details about the cows that were used for the 10 samples within the farms (e.g parity, days in milk, milk production) and then also show how many cows you sampled relative to the herd size. For example, you noted the average herd size was 86 – but what was the range in farm size? And thus what is the range in the percentage of the cows you sampled? Both of these aspects can be discussed as reasons why you did not find strong correlations between hair and welfare assessments.

How were cows chosen? This is severely lacking in the manuscript, this needs to be described in detail especially if you do not have information on the cows.

I also think that the results are presented a bit unfairly, for example in Table 4 there many assessments missing from the table that would fit under the classifications you chose and I’m not sure why. I think that you need report both null results and significant results. For example in your summaries in Section 3.5, you could have one sentence at the end of each paragraph that states what were not significant as well – just to make it more well-rounded.

You need to discuss that yes, the hair cortisol concentrations were not well correlated with the animal welfare assessments – but that the animal welfare assessment were not well correlated to each other either. This seems to be brought in the abstract and conclusions, but it is not included in the discussion and should be.

Finally, you need to discuss that the welfare assessments are broad and touch on many aspects of the farm. You mention that it was unexpected that the health sections do not correlate with the hair samples.  But now looking at the supplementary material, inside of the health section for WQ (just as an example), they include dehorning procedures. I think this concept need to be discussed fully or else this paper should be rejected.

L17 - “… in hair.”

L18 – “.. has been suggested to reflect stress levels over a long period of time.”

L26 – you list 9 assessements, not 8

L79 – “Obtained on farm”

L85 “… and is correlated with body condition score..”

L86 – “posture of the cow”

L88 = “.. stimulates milk production.”

L114 “ … or non-invasively, in saliva samples, reflect…”

L117 “a non-invasive”

L131 –“..answer the question of whether..”

L152 and 157 – I would suggest using the word “veterinarians” instead of “vets”.

L168 -169 – This sentence is a bit confusing. Do you mean 3 different observers went to each farm? Where each observer was performing a different protocol?

L171 – What percentage of the herd size was used? Include this information here such that the reader does not need to look in another document for information.

L173 – “.. result was..””

L179 – how were the cows selected for the animal-based observation? Was it the first cow the observer walked upon? Or there was some criteria?

L186-188 – Can you please include how the samples were stored over this year? Were they cleaned already? Were they ground?

L275 – This sentence isn’t complete, I think there was a mistake.

L286 – “…yielded low negative values…”

L286 – by “low negative” do you mean a negative value that is still close to zero? I would reword this, because -0.40 is lower than -0.16 – so this is kind of counter intuitive of what you are trying to say.

L290 – Maybe say the strongest correlation between cortisol and the welfare index? Highest would imply the most positive, when I believe you are trying to say the most negative.

L309 – Aren’t the rank correlations positive not negative?

Table 3 – The formatting is off.

Table 4 – Why do you no discuss tendencies within your results. For example Bedding Softness is -0.254 p=0.054 (so just over the limit for significance) – I think this is a very interesting result! Or there are some other tendencies that support your findings from other protocols that I think you could also bring up.

Table 4 – Why are there two headers with “A)”

Table 4 – I really like this table as it summarizes and combines all the protocols together in like aspects. But I am confused why you did not include all the health based measures from all of the protocols in this table. Honestly seems like the scores that are nearing or significant have been left out – maybe I don’t understand the table? Examples of missing scores: Welfare (KK), Mastitis (WW), Health (WQ), Behaviour (WQ), Production (WW). I know this data is a lot, there are a lot of small details and correlations but everything should be inclusive and fair.

L335  - Again here I think you miss explaining the whole picture. Why do you not compare the WQ (and derived) scores with the other protocols? Isn’t it more likely that scores that were derived from the same system would be correlated? And wouldn’t it thus be more novel to talk about how these scores are correlated with protocols that assess welfare in a completely different way. So for example, the WW End score, is not correlated with any of the protocols outside of the WQ and derived protocols, but this is not mentioned in the results.

L378-  were they left untouched until analyzed? This is the information I mention above that should be in the materials and methods (specifically for the ones that were left an extra 12 months).

L383 – You did not mention in the materials and methods that you did not enroll primiparous cows. If you do not have DIM or other information on the cows, please state this in the materials and methods to be transparent.

L384 – How can you assume the cows had similar DIM? How did you choose them? E.g. were they coming back from the milking parlour? We know there are different motivations dependant on DIM and milk production of who goes to the parlour first or last. If there were multiple groups in the barn did you select from each group? This information needs to be included in the material and methods.

L400 – You found a tendency for the KK – animal welfare section – do you not think this could be included as a discussion point here?

L412 – You reference for DIM and parity not impacting cortisol only looked at midlactation and only multiparous cows. There has been other research performed that would suggest both DIM and parity are important. This should be discussed.

Burnett, T.A., Madureira, A.M.L., Silper, B.F., Tahmasbi, A., Nadalin, A, Veira, D.M., and R.L.A. Cerri. 2015. Relationship of concentrations of cortsiol in hair with health, biomarkers in blood, and reproductive status in dairy cows.  J. Dairy Sci. 98:1-13.

Fukasawa, M., H. Tsukada, T. Kosako, and A. Yamada. 2008. Effect of lactation stage, season and parity on milk cortisol concentration

in Holstein cows. Livest. Sci. 113:280–284.

L417 – You did find -0.284 p= 0.34 for hair cortisol and the WW mastitis score but not for the KK subscores – so would you not conclude that your results are mixed? And not that they do not agree?

L422 – You did not mention scoring for lameness in the materials and methods – was this used as an exclusion criteria for the cows your hair sampled? Or can you report how many were lame?

L424 – As well you did not control for the health of the cows that you sampled – so it is not a surprise that you may not find strong correlations of 10% of your cows with the entire herd.

Supplementary material

Thank you for including the supplementary material – it makes your manuscript much stronger.

However, some time needs to be taken to go back through and make it stand alone so anyone can understand and generally be more clear.

Examples:

  • You need to use the same accronyms on the chart as in the text. Which was is KK? Which one is WQ??
  • BCS < 1.5 – what scoring system was used? This can just be included as a little footnote
  • “with thick hocks…. < a fist”? I believe you mean “swollen” instead of “thick”, and instead of just writing “< a fist” should be “< than size of a fist”
  • “Carpi”? Carpal joints?
  • “With arthritis” – how do you assess that a cow has arthritis?? Again this could be a footnote
  • “White dirty hind quarters” – I can not understand what 25x25 – 50x 50 cm means. Please write more clearly.
  • “with scabies” – I believe you mean “scabs”?
  • “with impaired teat condition” – is this referring to the teat end? Or just the entire teat?
  • Lameness – include the scoring system as a foodnote (if its not the same between welfare assessments state this.
  • “Width of the feeding place” - do you mean length of the feedbunk?
  • “Bedding is soft” – what does that mean? Can you add a few examples to be more well understood?
  • “Diagonal” -??
  • “Percentage dead cows” – per what? Per year?
  • “Non return > 56 days” – is this a percentage of cows? - adding the units to measures where it makes sense may save a lot of confusion.

I will not continue. But please go through, make it clear, where applicable add the scoring system as a footnote, add units when applicable. The section of Welfare Quality is much more clear you could use this as a “model”.

Author Response

Reviewer 1:

R1: The authors have improved this manuscript and added some very important details, but I still believe it is missing information to make it transparent and thus repeatable for other researchers. Overall it is still quite well written although the introduction could be more concise. There are many limitations to this study. These limitations should be clear to the reader (e.g. no control was taken for which cows were sampled interms of lactation status, health status, percentage of the cows sampled, welfare assessments are very broad etc).

>>> We thank the reviewer for the remarks and comments. We agree that the introduction is rather long, but we think that because of the complexity of the welfare assessment systems and the effect of stress on animal welfare, it is necessary to explain this here. The limitations of the study are mentioned and discussed in the revised manuscript.

R1: As mentioned in the last revision (and also by other reviewers as well), I think you need to state details about the cows that were used for the 10 samples within the farms (e.g parity, days in milk, milk production) and then also show how many cows you sampled relative to the herd size. For example, you noted the average herd size was 86 – but what was the range in farm size? And thus what is the range in the percentage of the cows you sampled? Both of these aspects can be discussed as reasons why you did not find strong correlations between hair and welfare assessments.

>>> As explained before, we do not have the data about parity, milk yield, and DIM. But even if we had those, which data would we use? The cortisol is the result of a 3 month period. In that period the yield of a cow changes. Cows can have been at the end of lactation, dry, and at the start of the next lactation in that period. Furthermore, we excluded primiparous cows because they could have been in another environment for a part of the 3 month period. The average parity in the Netherlands is around 2.3, so most of the sampled cows were in their second parity.
          Another important factor for cortisol levels in cattle is the social rank of each animal. In order to determine this, one needs a few days. And this has to be established at more times during a 3 month period because the group composition changes every time a cow enters or leaves the lactating group. This happens every few days. We do not have these data either. This might, however be an important factor causing the lack of correlation between cortisol levels and the welfare assessments. This is now included in the discussion.
          The number of animals per farm ranged from 27 – 224. This implies that the percentage of cows sampled differed per farm. The sample of small farms was relatively large in comparison with that of large farms. But at large farms the population is less diverse because the cows will be in all stages of lactation and parity. Therefore, one should take a relatively larger sample at a small farm.

R1: How were cows chosen? This is severely lacking in the manuscript, this needs to be described in detail especially if you do not have information on the cows.

>>> The cows were not ‘chosen’, but the first 10 that were used for the ‘clinical inspection’ were sampled. The WQ protocol has very specific instructions for this and they were followed. These were in practice cows that were systematically, randomly, sampled. The cows were at the feed rack and the first cow was randomly selected by choosing a number between 1-5. The total number of remaining cows was then divided by the number of cows that had to be sampled according to the table of WQ, resulting in X. Then every Xth cow was clinically examined and from the first 10 animals a hair sample was taken. But, as mentioned before, the procedure is extensively described in the WQ protocol and we refer to that publication. Cows do not have a specific place at the feed rack and this, therefore, can be considered the best option to randomly select animals from the herd.

R1: I also think that the results are presented a bit unfairly, for example in Table 4 there many assessments missing from the table that would fit under the classifications you chose and I’m not sure why. I think that you need report both null results and significant results. For example in your summaries in Section 3.5, you could have one sentence at the end of each paragraph that states what were not significant as well – just to make it more well-rounded.

>>> That would mean an enormous amount of data, because there were so many correlations calculated. We don’t think it will improve the readability of the paper, nor the clarity of the results.

R1: You need to discuss that yes, the hair cortisol concentrations were not well correlated with the animal welfare assessments – but that the animal welfare assessment were not well correlated to each other either. This seems to be brought in the abstract and conclusions, but it is not included in the discussion and should be.

>>> We agree and added this in the discussion.

R1: Finally, you need to discuss that the welfare assessments are broad and touch on many aspects of the farm. You mention that it was unexpected that the health sections do not correlate with the hair samples.  But now looking at the supplementary material, inside of the health section for WQ (just as an example), they include dehorning procedures. I think this concept need to be discussed fully or else this paper should be rejected.

>>> Indeed, the dehorning is included in the health section of WQ. This is done at a very young age and will not have a big impact on the stress level of lactating cows. This is why we choose to analyze correlations of the separate components of the criteria and principles of WQ (and the other assessment protocols). The discussion is extended to explain this in more detail.

R1: L17 - “… in hair.”

>>> Done

R1: L18 – “.. has been suggested to reflect stress levels over a long period of time.”

>>> Done

R1: L26 – you list 9 assessements, not 8

>>> Done

R1: L79 – “Obtained on farm”

 >>> Done

R1: L85 “… and is correlated with body condition score..”

>>> Done

R1: L86 – “posture of the cow”

>>> Done

R1: L88 = “.. stimulates milk production.”

>>> Done

R1: L114 “ … or non-invasively, in saliva samples, reflect…”

>>> Done

R1: L117 “a non-invasive”
>>> Done

R1: L131 –“..answer the question of whether..”

>>> Done

R1: L152 and 157 – I would suggest using the word “veterinarians” instead of “vets”.

>>> Done

R1: L168 -169 – This sentence is a bit confusing. Do you mean 3 different observers went to each farm? Where each observer was performing a different protocol?

>>> Yes indeed. The sentence is changed into: “One farm was visited by a different observer for each protocol (WQ, WW & KK).”

R1: L171 – What percentage of the herd size was used? Include this information here such that the reader does not need to look in another document for information.

>>> This is a very long table. (about one page). And since it does not have a major influence on the sample size for the present study, we think that it is not essential here. 

R1: L173 – “.. result was..””

>>> Done

R1: L179 – how were the cows selected for the animal-based observation? Was it the first cow the observer walked upon? Or there was some criteria?

>>> As explained above we followed the WQ protocol to select the animals for the animal based (clinical inspection) observations. This is now included in the text here.

R1: L186-188 – Can you please include how the samples were stored over this year? Were they cleaned already? Were they ground?

>>> They were not processed in any way. This is now included in the text.

R1: L275 – This sentence isn’t complete, I think there was a mistake.

>>> Indeed, the sentence is deleted.

R1: L286 – “…yielded low negative values…”

>>> Done

R1: L286 – by “low negative” do you mean a negative value that is still close to zero? I would reword this, because -0.40 is lower than -0.16 – so this is kind of counter intuitive of what you are trying to say.

>>> Indeed, this is confusing. We changed it into ‘small’. 

R1: L290 – Maybe say the strongest correlation between cortisol and the welfare index? Highest would imply the most positive, when I believe you are trying to say the most negative.

>>> Indeed, this is confusing. We changed it into ‘strongest’. 

R1: L309 – Aren’t the rank correlations positive not negative?

>>> They can be both positive or negative.

R1: Table 3 – The formatting is off.

>>> Not in the version I see on my screen.

R1: Table 4 – Why do you no discuss tendencies within your results. For example Bedding Softness is -0.254 p=0.054 (so just over the limit for significance) – I think this is a very interesting result! Or there are some other tendencies that support your findings from other protocols that I think you could also bring up.

>>> This is indeed an interesting finding that confirms the ideas we have about comfortable lying places. It was therefore already included in the discussion: Line 427-429: “The correlations with the freestall components and the overall assessment in the CCSS were low and nonsignificant, except for the softness of the bedding (ρ = -0.25; P=0.05). This is an important feature of a freestall and substantially influences the time spend lying [64-70].”

R1: Table 4 – Why are there two headers with “A)”

>>> This is an effect of the ‘track changes option’. The upper row is now deleted.

R1: Table 4 – I really like this table as it summarizes and combines all the protocols together in like aspects. But I am confused why you did not include all the health based measures from all of the protocols in this table. Honestly seems like the scores that are nearing or significant have been left out – maybe I don’t understand the table? Examples of missing scores: Welfare (KK), Mastitis (WW), Health (WQ), Behaviour (WQ), Production (WW). I know this data is a lot, there are a lot of small details and correlations but everything should be inclusive and fair.

>>> You are right to state that the amount of data is much. In fact it is too much, we had 212 variables in total. Therefore we choose to present only the ones that were relevant. As can be seen in the tables, we also presented correlations that had higher p levels. Furthermore, in other tables the requested correlations are presented (Table 2 has Welfare (KK), Table 3 Mastitis (WW), Health (WQ), Behaviour (WQ) and production (WW)).

R1: L335  - Again here I think you miss explaining the whole picture. Why do you not compare the WQ (and derived) scores with the other protocols? Isn’t it more likely that scores that were derived from the same system would be correlated? And wouldn’t it thus be more novel to talk about how these scores are correlated with protocols that assess welfare in a completely different way. So for example, the WW End score, is not correlated with any of the protocols outside of the WQ and derived protocols, but this is not mentioned in the results.

>>> It was not the intention of this study to compare these welfare assessment systems. However, it is inevitable that we calculated several correlations, especially at component-level. Our focus was directed to the relations with cortisol over a long period of time. To discuss all the correlations between the various assessments would exceed the intention of the current study. 

R1: L378-  were they left untouched until analyzed? This is the information I mention above that should be in the materials and methods (specifically for the ones that were left an extra 12 months).

>>> Yes, they were untouched, kept in a dry and dark place. This now mentioned in the M&M section. L 187-188: “The samples were stored in a dry and dark place at room temperature without any (pre)processing.”

R1: L383 – You did not mention in the materials and methods that you did not enroll primiparous cows. If you do not have DIM or other information on the cows, please state this in the materials and methods to be transparent.

>>> This was mentioned in L180-181 in the M&M section: “Heifers were excluded because it was not ascertained that they had been in the same environment for the past half year.” We added: “No data were collected about age, parity, milk yield, and days in milk.”

R1: L384 – How can you assume the cows had similar DIM? How did you choose them? E.g. were they coming back from the milking parlour? We know there are different motivations dependant on DIM and milk production of who goes to the parlour first or last. If there were multiple groups in the barn did you select from each group? This information needs to be included in the material and methods.

>>> There was no selection of the cows. The cows were not coming from the milking parlor, but were fixed at the feeding fence when the observer entered the barn. This all according to the WQ protocol, which is quite extensive about this procedure. The participating farms did not have different groups (age or production level). The latter is now included in the M&M section. (L 162-163)

R1: L400 – You found a tendency for the KK – animal welfare section – do you not think this could be included as a discussion point here?

>>> Yes, and so we did (L 406-410): “The welfare component of KK had a trending negative correlation (ρ = -0.23; P<0.085) with the hair cortisol level. This is an aggregation of several parameters: activity, BCS, locomotion, swollen hocks, hygiene, deviating cows, and general impression [12]. There was a fair correlation between the welfare component of KK and the EV (ρ = 0.44; P<0.001). The EV was based on some of these parameters”.  

R1: L412 – You reference for DIM and parity not impacting cortisol only looked at midlactation and only multiparous cows. There has been other research performed that would suggest both DIM and parity are important. This should be discussed.

>>> This part of the discussion is extended and a reference (Burnett et al. 2015) has been added. In this reference is stated that : “Within both experiments, a parity effect was present where multiparous animals consistently had higher cortisol concentrations than primiparous animals.” Since we excluded primiparous cows there should be no major problem.

Burnett, T.A., Madureira, A.M.L., Silper, B.F., Tahmasbi, A., Nadalin, A, Veira, D.M., and R.L.A. Cerri. 2015. Relationship of concentrations of cortsiol in hair with health, biomarkers in blood, and reproductive status in dairy cows.  J. Dairy Sci. 98:1-13.

Fukasawa, M., H. Tsukada, T. Kosako, and A. Yamada. 2008. Effect of lactation stage, season and parity on milk cortisol concentration in Holstein cows. Livest. Sci. 113:280–284.

>>> This paper is now included in the discussion (line 411-412). (was already mentioned in line 454)

R1: L417 – You did find -0.284 p= 0.34 for hair cortisol and the WW mastitis score but not for the KK subscores – so would you not conclude that your results are mixed? And not that they do not agree?

>>>  Point is here that mastitis lasts only for a few days. During those days there will be high cortisol levels. However, in a hair growth period of several months, you will not find this. Furthermore if she had a mastitis case 2 months before, one will find no symptoms during the assessment. The same is true for lameness. At the moment of assessment the cow may be lame, but this can exist for 1 day or 3 weeks (or in bad cases, months). Or a non-lame cow at the moment of assessment can have been lame for 2 months in the period of hair growth. The latter will have a high cortisol level in her hair and the score for lameness low.

R1: L422 – You did not mention scoring for lameness in the materials and methods – was this used as an exclusion criteria for the cows your hair sampled? Or can you report how many were lame?

>>> This certainly was not an exclusion criterion. The number of lame cows is published elsewhere (reference 11). https://dspace.library.uu.nl/handle/1874/372162 page 46 – 54.

R1: L424 – As well you did not control for the health of the cows that you sampled – so it is not a surprise that you may not find strong correlations of 10% of your cows with the entire herd.

>>> We did not sample 10% of the cows, but 10 cows per herd. And we did not exclude animals with health problems. These are measured during the animal based measurements.

R1: Supplementary material

Thank you for including the supplementary material – it makes your manuscript much stronger.

However, some time needs to be taken to go back through and make it stand alone so anyone can understand and generally be more clear.

>>> If I would give a full explanation of every assessment system here, the paper will become a full book. This table is meant to provide information as to what kind of information is gathered during an assessment. Not to provide detailed protocols. With the information in the tables one knows if the system is based on animal based or environment based measures. One has also an impression about the amount of measures taken. So the information I added is limited. I took the WQ table as example as suggested.

Examples:

  • You need to use the same accronyms on the chart as in the text. Which was is KK? Which one is WQ??
  • >>> These have been added to the tables
  • BCS < 1.5 – what scoring system was used? This can just be included as a little footnote
  • >>> Done
  • “with thick hocks…. < a fist”? I believe you mean “swollen” instead of “thick”, and instead of just writing “< a fist” should be “< than size of a fist”
  • >>> Done
  • “Carpi”? Carpal joints?
  • >>> Done
  • “With arthritis” – how do you assess that a cow has arthritis?? Again this could be a footnote
  • >>> Done
  • “White dirty hind quarters” – I can not understand what 25x25 – 50x 50 cm means. Please write more clearly.
  • >>> Done
  • “with scabies” – I believe you mean “scabs”?
  • >>> No we mean scabies, this is an infestation with a parasite that lives in the skin, causing heavy irritations.
  • “with impaired teat condition” – is this referring to the teat end? Or just the entire teat?
  • >>> Done
  • Lameness – include the scoring system as a foodnote (if its not the same between welfare assessments state this.
  • >>> Indeed, this differs among assessment systems but it becomes clear that there is a 1-5 point scale. It is mentioned now that this is according to the Sprecher system.
  • “Width of the feeding place” - do you mean length of the feedbunk?
  • >>> Yes, indeed. This is changed
  • “Bedding is soft” – what does that mean? Can you add a few examples to be more well understood?
  • >>> This is as it is mentioned in the description of the system. Indeed not very clear. But we interpret this as a non-painful knee-test
  • “Diagonal” -??

>>> this is the distance between the neck rail and the curb

  • “Percentage dead cows” – per what? Per year?

>>> Indeed, per year.

  • “Non return > 56 days” – is this a percentage of cows? - adding the units to measures where it makes sense may save a lot of confusion.

>>> The Non return < 56 days is indeed a percentage of cows. It is a well-known index for the success rate of inseminations and is the percentage of cows that is not inseminated again within 56 days after insemination.

More additions have been made

Reviewer 2 Report

The relation between hair-cortisol concentration and various welfare assessments of Dutch dairy farms

Dear authors,

Extensive changes have been made to the above-mentioned manuscript. 

I am satisfied with the changes made and the manuscript is now improved highly to the standards required by the journal.

The only minor issue that I am not sure if the journal editors have considered is that having 80 references for a research article is too many. I usually ask authors to reduce the references to be around 40-50 otherwise, it would be considered as a review paper. So, if the same issue raised by the editor, please reduce the number of references.

Author Response

Dear authors,

Extensive changes have been made to the above-mentioned manuscript. 

I am satisfied with the changes made and the manuscript is now improved highly to the standards required by the journal.

>>> Thank you very much for your positive comments.

The only minor issue that I am not sure if the journal editors have considered is that having 80 references for a research article is too many. I usually ask authors to reduce the references to be around 40-50 otherwise, it would be considered as a review paper. So, if the same issue raised by the editor, please reduce the number of references.

>>> We understand that it is indeed a long list, but many issues are addressed here and the matter is rather complex. So we think that we need all these references. The editor did not comment on this.

Round 2

Reviewer 1 Report

I believe the authors had now included enough information within the paper that I can now show all of its strengths and weaknesses equally, which is great. Thank you for the effort you put into the supplementary material. I believe it is sufficient now to be published.

I do not agree with the comments of the authors from that last review that you would have an issue defining DIM someway in the analysis – as an example, as you mention, you could have cows where calving occurred within the last three months, or you could have cows that have been late lactation low producing cows for the last three month. Without collecting this data there is no way for you to know and you can only assume that your sampling technique was equal across farms – which is very doubtful since on your smallest farm you sampled 37% of the cows while on your largest farm you sampled less than 5% of the cows.  This applies to all other parameters that you mentioned as well (parity, social rank etc.). That being said, this information is now within the text so the readers can make their own judgements on the matter. Also, thank you for including how the 10 cows were randomly included, this is very helpful information, and was not clear in previous versions.

L309- 311 – These correlations are supposed to be positive I believe. My last comment was not asking if correlations can be positive and negative – it was to say that I believe there is an error. In the table these correlations are positive, while the values are written as negative here.

L477 – L481 Please include this statement within the discussion of the paper as well. You should not bring up a new topic within the conclusions.  It was noted by the authors that this was included in the discussion, but I could not see its addition there. Maybe there was a mistake.

Author Response

I believe the authors had now included enough information within the paper that I can now show all of its strengths and weaknesses equally, which is great. Thank you for the effort you put into the supplementary material. I believe it is sufficient now to be published.

>>> We thank the reviewer for the efforts and time spent with our manuscript.

I do not agree with the comments of the authors from that last review that you would have an issue defining DIM someway in the analysis – as an example, as you mention, you could have cows where calving occurred within the last three months, or you could have cows that have been late lactation low producing cows for the last three month. Without collecting this data there is no way for you to know and you can only assume that your sampling technique was equal across farms – which is very doubtful since on your smallest farm you sampled 37% of the cows while on your largest farm you sampled less than 5% of the cows.  This applies to all other parameters that you mentioned as well (parity, social rank etc.). That being said, this information is now within the text so the readers can make their own judgements on the matter. Also, thank you for including how the 10 cows were randomly included, this is very helpful information, and was not clear in previous versions.

>>> The information is indeed in the text now and everybody can make their own judgements.

L309- 311 – These correlations are supposed to be positive I believe. My last comment was not asking if correlations can be positive and negative – it was to say that I believe there is an error. In the table these correlations are positive, while the values are written as negative here.

>>> In the current version both are negative in the table 3 and text (line 309-310).

L477 – L481 Please include this statement within the discussion of the paper as well. You should not bring up a new topic within the conclusions.  It was noted by the authors that this was included in the discussion, but I could not see its addition there. Maybe there was a mistake.

>>> This was already mentioned in the text at line 396-397. Now we have added here: ‘corroborating earlier findings [17]’.

This manuscript is a resubmission of an earlier submission. The following is a list of the peer review reports and author responses from that submission.

Round 1

Reviewer 1 Report

This paper describes the associations of welfare assessments and hair cortisol concentrations as a measure of chronic stress. I believe this manuscript is important for the body of knowledge about hair cortisol and is appropriate for the journal. This paper was extremely well written. I have left some major and minor comments below.

I think this manuscript is lacking on transparency and detailed explanation of what exactly was measured. Further description of what exactly the farm assessments were (which animals were used? The entire farm? Subset?, over what period of time, what behaviours and illnesses were included etc).

Similarly, how were the 10 cows picked on each farm? Do we know if this is a representative sample of each farm? Or is it more of a convenience sample? Cows are known to have varying hair cortisol concentration throughout their lactation ( eg. early laction vs late lactation), as well as health indicators such as lameness, mastitis, metritis etc are also strong correlated with days in milk.

This difference in timing and stage of lactation of the animals could be a reason explaining why you are not finding strong correlations between the herd level (using 10 cows) hair cortisol against a farm level assessment of welfare that may be including a different set of animals (or those in a different stay of lactation). This may also attribute to why you did find an association with vet opinions and hair cortisol concentrations.

Better descriptions of these details are needed.

L50-69 – I understand that most of the assessments are based off of the 5 freedoms. But could you mention how many of the assessments include an assessment of the animal themselves (e.g.: hock scores, lameness, standing time etc.), or are these assessments all based on management and the environment?

L72 “.. posture of the cow.”

L112-113 – These papers cited are not welfare assessments – they are just methodologies for measuring stress.

L193 – Ground not grinded

L256 – the sign is wrong for one of the + 0.2 to +0.3

L291 – not all your significant results are highlighted. Eg. Average cortisol and vet estimates.

L291 – In the text you say the highest correlation between cortisol and the assements was with the vets, but the WQ adjusted has a higher correlation (0.43 vs -0.28). Why?

L295 – Table 2. Better descriptions of what you mean by “Feeding”, “Housing”, “Health” etc need to be included either as a footnote or within the text. You could also contemplate including an appendix with summary tables of what is measured in each of the assessments.

Table 3 – if you are going to highlight in some tables please highlight in all.

Table 2 and 3 – When you speak to lameness, mastitis etc. Is this on the herd level? Or from the cows you sampled? Please clarify in the table titles.

L314 – are there any differences between these papers on how they are stored? For example were some ground, some cleaned? Or were the left after collection without any processing?

Reviewer 2 Report

 Article 1

The relation between hair-cortisol concentration and various welfare assessments of Dutch dairy farms

This manuscript is well written. The objective is clear and has novelty. A high number of animals is a strong part of this work which should be appreciated due to helping the power of analysis. However, there are major issues including the wide area of the animal’s body which hair samples were harvested as shown in Fig.1. Although even we ignore peripheral HPA-axis activity, which we should do so because its effect is still unclear, since the main source of hair cortisol is via blood interfering with the hair cells, blood circulation and regional choice may affect the hair cortisol concentration (as stated by some literature). Thus, this measurement could be only valid if all hair samples were taken from the exact same region of the body and from the same animals. The range of age, stage of lactation, and body weight of animals are also not defined. Moreover, although the conclusion is supported by the obtained results, however, due to the aforementioned reason, the obtained results are questionable.  Furthermore, reading the “Collecting hair samples” in M&M section, it appeared that not only the wide area of flank (not exactly the same spot) was subjected to hair sampling, but also authors did not define that whether the first hair cut was subjected to hair cortisol? This can show only the previous accumulation of cortisol into the hair shaft but not the duration of the experimental period. Only new hair grown cut after first time cutting can be used for hair sample analysis. While the age and body weight plus the physiological status of the animals (parity and lactation stage) can affect hair cortisol concentration, these effects were ignored in this study. Given these, the wide range of hair cortisol concentration as stated in line 261 is inevitable and not thus accurate for correlation analysis.

Collectively, despite the large sample size in this study, the results of this study are not valid for evaluation.

The Introduction is too long and there are unnecessary statements in the Introduction such as line 70-80 in the 2nd paragraph of Introduction. Though it introduces another index for evaluating the welfare status of the cows, it has nothing to do with the objective of this study which mainly discussing hair cortisol assessment. Another issue is that also in the 2nd paragraph authors cited two reliable references; however, milk yield cannot be a single indicator of welfare where its increase and decrease both can and cannot be associated with the welfare of the cows. The cows can provide less welfare and having high milk yield and vice versa. The number of references is too much and usually, for a research article, more than 40 references are not allowed. Also, some related references are missed.

Reviewer 3 Report

Review animals-946450

The manuscript covers an important and current topic with the combination of hair cortisol measurement of assessing chronical stress situations and the validity of several welfare assessment schemes. The difficulty is, that both the welfare schemes and the hair cortisol measurement are complex and can be influenced by many factors. Therefore, both can be discussed critically and both are not ‘hard figures’. Thus, some conclusions must be drawn more carefully. Nevertheless, there seem to be some methological weaknesses or at least lack of explanations. These aspects should be clarified or and discussed better. Details are listed below:

line 26-29, 54-75, 137, 164-169: The number of considered welfare assessment schemes is big, the number of listed schemes varies between 6, 8 and 9 schemes in the descriptions. This is quite confusing, also because the names of the schemes are partly quite similar. I suggest to mention at all paragraphs the number of schemes and perhaps use a kind of numbering for example A, B, C or (1), (2), (3),.... Additionally, it would be very helpful for the reader to provide a table with information about contents and indicators of the different welfare assessment schemes: e.g. with lines of all animal-based and resource-based indicators and columns of the different welfare schemes that contains crosses of those indicators that are used in the specific scheme. This table may also contain the specific aim of the assessment scheme (it seems to be a wide range of more complex schemes of e.g. WQ and more single indicators like SSI. It would be very helpful to get an easier overview about these schemes than provides currently in the text.

Line 34-37: these information are not easy to understand without reading the whole manuscript.

line 51: reference to the WQ cattle protocol might be added.

Line 55: You mention correlations between the welfare assessment schemes. Consequently, please provide also those results.

Line 115-116: In faeces and urine, cortisol metabolites are analyses and not cortisol itself. Please rephrase with more exactly by adding ‘metabolites’, also at the other places were fecal cortisol metabolites are mentioned.

Line 116: ‘both’ is perhaps not the appropriate word, because you combine saliva, faeces and urine in one term. For saliva it is correct, that it reflects acute stress. Urine is normally not used in cattle, but fecal cortisol metabolites are not only reflecting acute stress but a period of several hours, therefore this indicators has also used to measure mid-term stress.

Line 123: further methological studies like Burnett et al regarding hair color and body region, Vesel et al 2020, Salaberger et al 2016, Gonzalez-de-la-Vara 2011 might complete the picture.

line 150,157: These >5 vets were from one vet practice? You ask all of them? Did all vets know each farm equally well? Please explain this procedure a bit clearer.

line 164: Please provide figures of an inter-observer reliability tests of the seven observers.

line 164: Who were those observers? The vets mentioned in line 150? If so, please discuss, that they might be pre-influenced by their previous impressions of the farms, leading to their classification of the farms.

line 164-169: Why are Welfare Index and CWM missing here?

line 172-173: Was it always possible to get white hairs in this body region? Please provide figures/percentage, if not. Please provide additional information about the breeds of the investigated farms. Did all have Holstein breeds? If not, I guess it would have been difficult to get always white hair?

Please explain ‚clean’. Did you wash the squares if the animals were dirty in this body region?

line 174: Please provide information about range of lactation stage e.g. by days in milk. Did you prove that this range was comparable for all farm? Did you prove the influence of days in milk or other animal characteristics on hair cortisol? Several factors can influence the hair cortisol, beside distress the values could also be elevated due to increased activity for example in the beginning of pasture season. Therefore, the selection of the 10 cows per farms should be described as detailed as possible and the impact of this cow selection should be discussed very critical in the discussion chapter. Additionally, please provide information, how many of the cows were clinically healthy, because this might also influence the results (you might have figures about this at least regarding severe lameness and integument lesions from the WQ protocol).

line 204: ‚Samples of the different farms were measured in duplicate’ is phrased unclear inmy opinion. This sounds like animal-pooled samples per farm? Previously I understood that you have measured the samples on animal level. Please specify.

line 244: I guess this sentence is incomplete?

line 261-262: You used mean and not median – were the data normally distributed? Please add in the 2.7). Due to possible outliers and impacts of single cows within farm it might be more reasonable to calculate medians instead of means to get the farm values?

line 269: incomplete sentence.

Line 278-280: this appears confusing: in line 278 you introduce correlations between welfare schemes and hair cortisol, but in line 279-280 you report a correlation between one welfare scheme and the vets’ preclassification. I would recommend to sort the results better into correlations between welfare assessments (and it is the question, if the vets’ preclassification is one; in any case this should be described in the material&methods chapter) and correlations between welfare schemes and hair cortisol.

Table 3: It is not in the sense of QBA to use the single terms, but only the first or even also the second dimension of the PCA.

Furthermore, many more details of the analyzed indicators visible here, should be explained in the text of material & methods.

Line 318: I do not agree with this sentence: I guess it is less a question of objectivity (welfare schemes are complex by nature, but objectivity has already been proven e.g. by inter- and intra-observer reliability of those assessments; here is a weakness of your study that you didn’t provide information about this) but more a question of complexity, and this refers to both to the welfare assessment schemes and to the hair cortisol measurement. The hair cortisol might be appearing more objective, because it is one figure, but the interpretation is not easy because positive activity can influence the cortisol level in the same direction than distress. And a lot of factors like metabolic activity, age, body region, hair color etc. can influence the results. Thus, some authors discuss the hair cortisol method critically, see e.g. Salaberger et al. 2016. Therefore, you study design should be discussed more critically. This comment refers also in line 17 and 31. In line 17 the word ‘objectively’ should be deleted.

Line 326-328: If this was not the intension to investigate, why did you include this factor in our analyses? I would suggest either to exclude it or include it from the beginning on in the aim and all chapters of the manuscript.

Line 329-332: This result again points on the need of better explain the cow selection per farm regarding comparability of days in milk. It might be discussed also in the context of elevated cortisol values by metabolic stress.

Line 337: perhaps the results vary, but there are results in other studies, that lactation number/age / rank and lactation stage influence cortisol values. (e.g. Gonzalez-de-la-Vara et al. 2011; Fukasawa et al 2008, Ebinghaus et al 2020)  (you mention it yourself in line 370-371. )

Line 355: see my comment above regarding use of single QBA terms.

Line 360: Due to complexity also of hair cortisol and due to methological weaknesses of the presented study, this conclusion is not valid from this study.

Line 372-374: Again, I would recommend to work with medians on herd level instead of means. You cannot change the animal selection retrospectively, but you might calculate the farm values again.

Line 383-384: this should be mentioned earlier and considered regarding calculation of farm values.

Discussion chapter in general: An aspect of study design regarding the statistics is lacking: You have calculated many single correlation tests. This would require a Bonferroni correction or at least a critical discussion.

Line 395-401: please phrase more carefully because you have methological weaknesses in your study design. Especially line your study design allows not the conclusions from line 397-401. (this refers also to line 21-23 and line 37-39).